# An operationally implementable model for predicting the effects of an infectious disease on a comprehensive regional healthcare system

**Daniel Chertok**[1]*, **Chad Konchak**[1], **Nirav Shah**[1,2], **Kamaljit Singh**[1], **Loretta Au**[1], **Jared Hammernik**[1], **Brian Murray**[1], **Anthony Solomonides**[1], **Ernest Wang**[1], **Lakshmi Halasyamani**[1,2]

**1** NorthShore University HealthSystem, Evanston, Illinois, United States of America, **2** University of Chicago Pritzker School of Medicine, Chicago, Illinois, United States of America

* dchertok@northshore.org

**Data Availability Statement:** All relevant data are within the manuscript and its Supporting information files.

## Abstract

An operationally implementable predictive model has been developed to forecast the number of COVID-19 infections in the patient population, hospital floor and ICU censuses, ventilator and related supply chain demand. The model is intended for clinical, operational, financial and supply chain leaders and executives of a comprehensive healthcare system responsible for making decisions that depend on epidemiological contingencies. This paper describes the model that was implemented at NorthShore University HealthSystem and is applicable to any communicable disease whose risk of reinfection for the duration of the pandemic is negligible.

## Introduction

Upon its emergence in 2020, the COVID-19 pandemic presented immediate challenges to the operation of NorthShore University HealthSystem (NS), a comprehensive regional healthcare system located in the northern part of Chicago, Illinois, and its suburbs. The need to forecast the expected demand on floor and ICU beds, ventilators and requisite supplies became pressing at the onset of the disease. The lack of reliable population data posed additional difficulty in implementing a usable model. Additional constraints of robustness, distributability and transparency imposed further requirements on the choice of the governing equations, solution algorithm and software implementation. During the initial stage of the pandemic, the model was delivered to the operational stakeholders daily; as time progressed, the frequency of dissemination was changed to once or twice a week, depending on the severity of the situation.

At the onset of the pandemic, the Clinical Analytics team was tasked with providing a reliable, scalable solution relevant to the local epidemiological situation [1]. While abundant literature exists on the theoretical aspects of the problem [2–21], few specific worked examples that could be used by practitioners for immediate implementation are widely available. For a

**Funding:** The authors received no specific funding for their work.

**Competing interests:** The authors have declared that no competing interests exist.

concise but comprehensive description of challenges facing a researcher attempting to develop a workable model see, e.g., [22]. While most of the existing publications that offer applicable practical solutions focus on country-wide statistics [23, 24], those dealing with local conditions are scarce. In order to find a satisfactory answer to this challenge, we had to quickly construct a flexible, scalable model easily adaptable to rapidly changing conditions that could be quickly communicated to a growing number of stakeholders while affording them an opportunity to create area-specific "back-of-an-envelope" analyses suitable for their needs. This task was accomplished by augmenting the industry-standard Susceptible-Infected-Recovered (SIR) model [2] with bootstrapping estimates and interpolation and extrapolation approximations of patient flow dynamics. The resulting model was robust enough to exhibit accuracy sufficient for predicting floor, general intensive care unit (ICU), ventilator census and mortality up to two weeks in advance. The main accomplishments of the foregoing approach were the ability to quickly adapt the model to the observed coefficient of transmission ($R_0$) prevalent in the hospital service area, compute the forward expected length of stay on the hospital floor, in the ICU and on the ventilator, and incorporate actual and projected vaccination rates into the model.

The paper is organized as follows. First, we review generally accepted modeling principles for forecasting the progression of the disease (COVID-19). Next, we provide empirical formulas for approximating dynamically observed rates of hospitalizations, ICU and ventilator placement, mortality and vaccination. Following that, we present the results and discuss their accuracy. We conclude with a summary of findings and directions for further research.

## Materials and methods

### General equations

The choice of a model was informed by the requirements of specificity to the available NS population data, ease of implementation through a common tool understood by operational leaders and ease of explanation to a non-mathematical audience. In general, available models meeting those criteria include variants of the SIR model [2, 3] with a time-dependent coefficient of transmission [4] and vaccination effects [5]. While a stochastic SIR model [6] presents a viable enhancement, estimating the parameters of the stochastic component may prove problematic during the onset of the pandemic. A plausible alternative, an Individual-Based Model (IBM) [7], requires substantially more effort devoted to implementation and analysis [8], and is more difficult to explain to the target audience than SIR.

From the practical standpoint, the need to develop a workable model prior to the publication of [4], as well as the need to have a robust, distributable software solution, necessitated the adoption of a simplified time-dependent form (note that the time dependency of $\beta$ precludes the use of an analytical solution described in [9]).

$$dS(t) = -(\beta(t)S(t)I(t) + V(t))dt , \tag{1}$$

$$dI(t) = (\beta(t)S(t) - \gamma)I(t)dt , \tag{2}$$

where

$S(t)$—fraction of the population susceptible to the disease,

$I(t)$—fraction of the population currently infected with the disease,

$\beta(t)$—coefficient of transmission,

$V(t)$—fraction of the population that has been vaccinated,

$\gamma$—fraction of the infected population removed from further consideration due to (permanent) recovery or death.

We are not concerned with the dynamics of the recovered population, and hence leave the "R" term out of Eqs (1) and (2). Since the dynamics for $\beta(t)$ and $V(t)$ are not known beforehand, they are not included in the differential equations as separate terms, and are instead left to be determined at a later discretization stage. For simplicity, we disregarded mobility considerations reviewed in [25, 26], as well as implemented barriers to transmission (masking, social distancing and quarantine measures) [27].

## Numerical solution

**Data extrapolation and scenario analysis.**   At the onset of the pandemic, there is no reliable way to determine the true number of infected patients and hence the transmission coefficient $\beta(t)$. While initial attempts were made to infer likely epidemiological dynamic from countries where the initial stage had by that time already passed [10], the validity of this approach was questionable even at that time since different locales exhibited different curve characteristics. In view of this, the approach adopted for the purpose of constructing a robust model applicable to local conditions was as follows:

1. assume that the number of observed NS cases reflected the actual count of the disease in the population. While this was certainly not the case initially, the accuracy of that number increase over time as testing became more prevalent and comprehensive; moreover, it is fair to assume that those inaccurate numbers reflected the qualitative dynamic of the pandemic;

2. extrapolate the evolution of $\beta(t)$ implied by the historical data (initially, piecewise-constant; subsequently, polynomial or 7-day moving average);

3. repeat 1–2 for the Chicago / Cook / Lake county (CCL) area containing the majority of the NS catchment area;

4. construct a dynamic (time-dependent) ratio of NS to CCL cases and assume that it accurately reflects the proportion of CCL patients attributable to NS;

5. solve Eqs (1) and (2) separately for CCL and NS;

6. use the minimum and maximum case number estimates from step 5 as boundaries for the expected number of NS cases.

Specifically, for step 2, we need to find the value of $\beta(t)$ that delivers an exact solution to (1) and (2) at $t$ (more on this in subsection below). We can do this by equating the number of newly discovered cases in the NS population less the number of those newly vaccinated to the instantaneous decline in the susceptible population (since the latter is monotonically decreasing, i.e., $dS(t) < 0$, $-dS(t) > 0$ represents the number of patients who have been infected or vaccinated at time $t$):

$$dI_+(t) = \lim_{\Delta t \to 0} I(t + \Delta t) - I(t) = -dS(t) - dV(t) \,. \tag{3}$$

Eq (3) applies to the CCL population as well.

## Numerical solution of the SIR equations

Conventionally, (1 and 2) are solved numerically using the 4-th order explicit Runge-Kutta method [28]:

$$k_1 = -(\beta(t)S(t)I(t) + V(t)) \,, \tag{4}$$

$$l_1 = (\beta(t)S(t) - \gamma)I(t) \,, \tag{5}$$

$$k_2 = -\left[\beta(t)\left(S(t) + \frac{h}{2}k_1\right)\left(I(t) + \frac{h}{2}l_1\right) + V(t)\right], \tag{6}$$

$$l_2 = \left[\beta(t)\left(S(t) + \frac{h}{2}k_1\right) - \gamma\right]\left[I(t) + \frac{h}{2}l_1\right], \tag{7}$$

$$k_3 = -\left[\beta(t)\left(S(t) + \frac{h}{2}k_2\right)\left(I(t) + \frac{h}{2}l_2\right) + V(t)\right], \tag{8}$$

$$l_3 = \left[\beta(t)\left(S(t) + \frac{h}{2}k_2\right) - \gamma\right]\left[I(t) + \frac{h}{2}l_2\right], \tag{9}$$

$$k_4 = -[\beta(t)(S(t) + hk_3)(I(t) + hl_3) + V(t)], \tag{10}$$

$$l_4 = [\beta(t)(S(t) + hk_3) - \gamma][I(t) + hl_3], \tag{11}$$

$$S(t + h) = S(t) + \frac{h}{6}[k_1 + 2k_2 + 2k_3 + k_4], \tag{12}$$

$$I(t + h) = I(t) + \frac{h}{6}[l_1 + 2l_2 + 2l_3 + l_4]. \tag{13}$$

The Runge-Kutta method (4–13) is explicit and therefore inherently unstable, however, it is conventionally applied for $h = 1$. The justification of this can be found, e.g., in [29].

## Estimating the number of potential NS patients

Eqs (1) and (2) are written in terms of population percentages, i.e.,

$$S(t) = \frac{\mathcal{S}(t)}{N(t)}, \tag{14}$$

$$I(t) = \frac{\mathcal{I}(t)}{N(t)}, \tag{15}$$

$$V(t) = \frac{\mathcal{V}(t)}{N(t)}, \tag{16}$$

where
$\mathcal{S}(t)$—NS population susceptible to the disease,
$\mathcal{I}(t)$—NS population currently infected with the disease,
$\mathcal{V}(t)$—NS population that has been vaccinated,
$N(t)$—NS population at time $t$.
In order to estimate $N(t)$, we assumed that the current proportion of NS cases relative to the observed CCL cases is indicative of the fraction of the CCL population that potential NS

patients represent. In other words,

$$\frac{\Delta \mathcal{I}_{+\mathcal{NS}}(t)}{\Delta I_{+CCL}(t)} = \frac{N(t)}{N_{CCL}} \ , \tag{17}$$

where

$\Delta \mathcal{I}_{+NS}(t)$—newly discovered NS cases at time $t$,

$\Delta \mathcal{I}_{+CCL}(t)$—newly discovered CCL cases at time $t$,

$N_{CCL}$—CCL population (deemed constant).

The left hand side of (17) is time-dependent. This seemingly contradicts the static assumption for $N$ implied by the form of (1 and 2). One could, at least partially, refute this objection by pointing out that $N$ is an *estimate at time t* of the true NS population. The exact number of potential NS patients in the NS catchment area is unknown since potential future NS patients may have had no prior contact with NS facilities and, conversely, those who have sought treatment at NS in the past may chose an alternative provide for their emergency care and subsequent recovery. It is presumed to approach the exact (steady-state) value asymptotically. The rationale for that is mostly empirical, however, one could hypothesize that, as the epidemic progresses, patient flow distributions across healthcare providers tend to stabilize. In fact, as can be inferred from Fig 1, empirical data derived from positive test results and patient censuses in various parts of the hospital suggests exponential decay of $\frac{N}{N_{CCL}}$ that can be approximated by

$$\frac{N}{N_{CCL}}(t) = \alpha + \beta e^{\mu t} \ , \tag{18}$$

where

$\alpha > 0, \beta > 0, \mu < 0$—empirically determined constants: $\alpha = 0.452, \beta = 1.61, \mu = -0.012$ (the presented empirical fit was performed on the data between Oct. 31, 2020 and May 30, 2021).

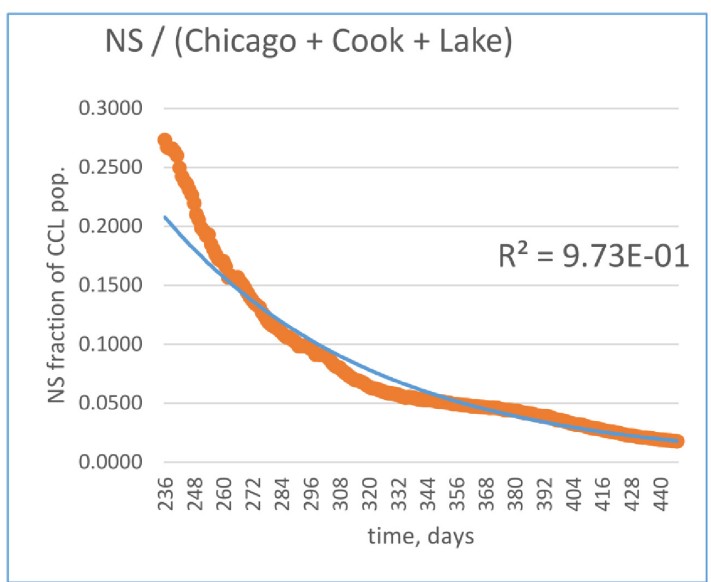

**Fig 1. The ratio of NS and CCL populations.** *x-axis*: elapsed time in days since March 10, 2020 (first identified NS case); *y-axis*: NS / CCL population ratio less 0.452 (asymptotic limit).

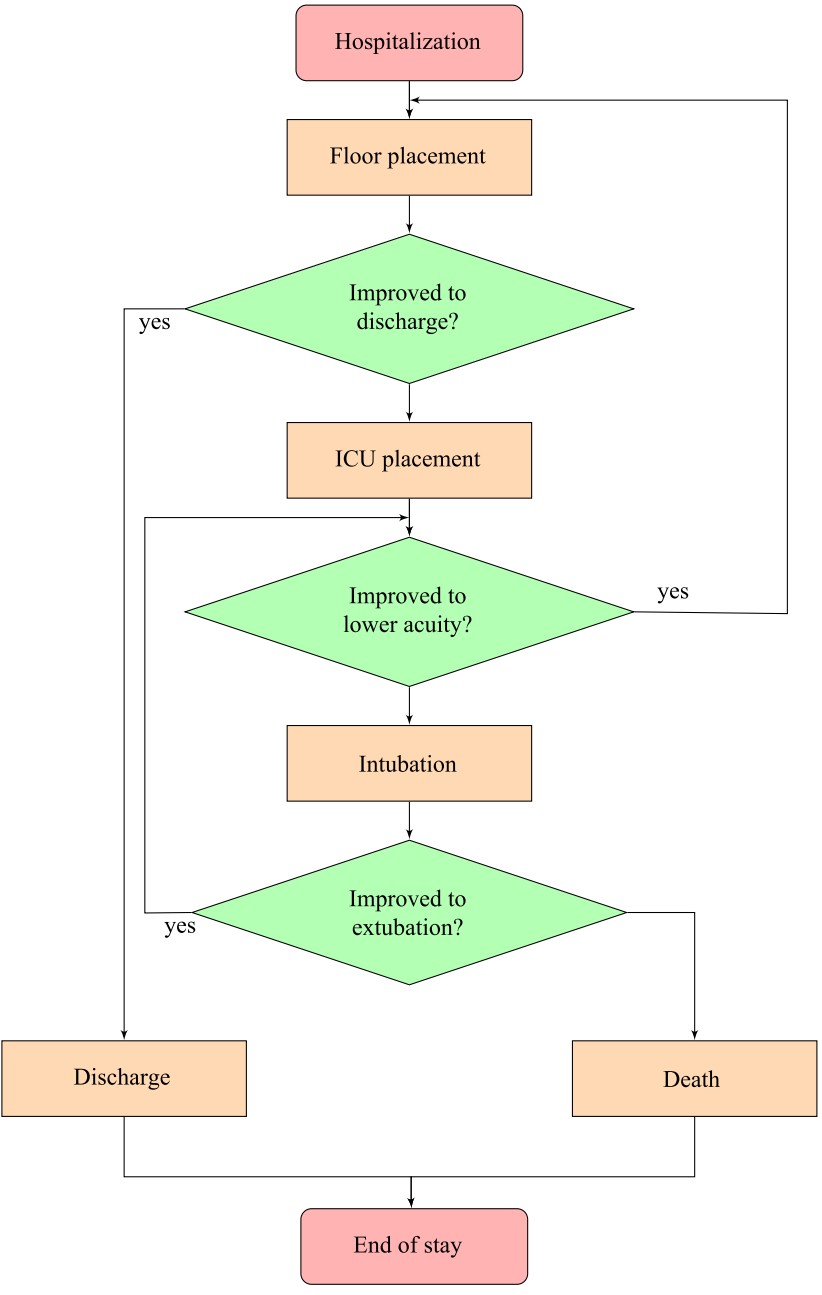

**Fig 2. Progression of a hospitalized patient through their stay.**

U.S. Census Bureau [30] estimates the population of the CCL area to be 5,846,768 residents as of July 2019 (the latest data available at the time of writing). From (18), we can obtain the population estimate for NS to be approximately 392,000.

## Projecting the number of hospitalizations, ICU and vent placements and deaths

In order to forecast the number of patients requiring general beds, ICU placement or intubation, we assume that, at any given time, a patient can be observed in any of the following states:

- on the floor but not in the ICU (and not intubated; lower acuity);

- in the ICU but not intubated (elevated acuity),

- intubated (highest acuity).

The flowchart in Fig 2 represents the progression of a hospitalized patient through his or her stay in the hospital. The following simplifying assumptions have been made:

- floor (lower acuity), ICU and intubated patients are accounted for separately, i.e., those are mutually exclusive groups;

- a patient is initially placed on the floor. If their condition is grave, transfer to ICU and / or intubation occurs (almost) instantly;

- upon deterioration, a patient proceeds from the floor to the ICU to intubation. No stages in this sequence are skipped, but a patient can spend almost no time in any state and be transferred to a higher acuity stage instantly (in other words, if a severely ill patient expires without being transferred to the ICU and / or being intubated, we consider that patient to have instantaneously transitioned through those two stages to mortality);

- upon improvement, a patient proceeds from intubation to the ICU to the floor as applicable. No stages in this sequence are skipped but a patient can spend almost no time and be transferred to a lower acuity stage instantly;

- there is no formal restriction on how many times a patient can deteriorate or improve.

Under those assumptions, the state equations describing the population dynamics inside the hospital are

$$
H_F(t) = \begin{cases} \dfrac{\mathcal{H}_{+F}(t; N_F)}{\mathcal{I}_{+NS}(t; N_F)} = \dfrac{\displaystyle\sum_{i=0}^{N_F-1}\Delta\mathcal{H}_{+F}(t_i)}{\displaystyle\sum_{i=0}^{N_F-1}\Delta\mathcal{I}_{+NS}(t_i)}, & T_0 <= t_0 < t_{N_F-1} <= T, \\[2em] \dfrac{1}{N_F}\displaystyle\sum_{i=0}^{N_F-1} H_F(t_i), & t_{N_F-1} > T \end{cases} \tag{19}
$$

$$
H_{ICU}(t) = \begin{cases} \dfrac{\mathcal{H}_{+ICU}(t; N_{ICU})}{\mathcal{H}_{+F}(t; N_{ICU})} = \dfrac{\displaystyle\sum_{i=0}^{N_{ICU}-1}\Delta\mathcal{H}_{+ICU}(t_i)}{\displaystyle\sum_{i=0}^{N_{ICU}-1}\Delta\mathcal{H}_{+F}(t_i)}, & T_0 <= t_0 < t_{N_{ICU}-1} <= T, \\[2em] \dfrac{1}{N_{ICU}}\displaystyle\sum_{i=0}^{N_{ICU}-1} H_{ICU}(t_i), & t_{N_{ICU}-1} > T \end{cases} \tag{20}
$$

$$H_{vent}(t) = \begin{cases} \dfrac{\mathcal{H}_{+vent}(t; N_{vent})}{\mathcal{H}_{+ICU}(t; N_{vent})} = \dfrac{\sum_{i=0}^{N_{vent}-1} \Delta\mathcal{H}_{+vent}(t_i)}{\sum_{i=0}^{N_{vent}-1} \Delta\mathcal{H}_{+ICU}(t_i)}, & T_0 <= t_0 < t_{N_{vent}-1} <= T, \\[4ex] \dfrac{1}{N_{vent}} \sum_{i=0}^{N_{vent}-1} H_{vent}(t_i), & t_{N_{vent}-1} > T \end{cases} \quad , \qquad (21)$$

$$t_i = t_{i-1} + \Delta t \ .$$

where

$N_F$—length of lookback period for floor patients (at the time of this writing, 14 days),

$N_{ICU}$—length of lookback period for ICU patients (14 days),

$N_{vent}$—length of lookback period for ventilated patients (14 days),

$H_F(t)$—hospitalization rate at time $t$,

$\mathcal{H}_{+F}(t; N_F)$—total number of new patients placed on the floor during the lookback period $N_F$,

$\mathcal{I}_{+NS}(t; N_F)$—total number of new infections identified among NS patients during lookback period $N_F$,

$H_{ICU}(t)$—ICU placement rate at time $t$,

$\mathcal{H}_{+ICU}(t; N_{ICU})$—total number of new patients placed in the ICU during lookback period $N_{ICU}$,

$\mathcal{H}_{+F}(t; N_{ICU})$—total number of new patients placed on the floor during the lookback period $N_{ICU}$,

$H_{vent}(t)$—intubation rate at time $t$,

$\mathcal{H}_{+vent}(t; N_{vent})$—total number of new patients placed on the ventilator during lookback period $N_{vent}$,

$\mathcal{H}_{+ICU}(t; N_{vent})$—total number of new patients placed on the floor during the lookback period $N_{vent}$,

$\Delta\mathcal{H}_{+F}(t)$—number of new patients placed on the floor at time $t$,

$\Delta\mathcal{H}_{+ICU}(t)$—number of new patients placed in the ICU at time $t$,

$\Delta\mathcal{H}_{+vent}(t)$—number of new intubations at time $t$,

$T_0$—time of the start of the pandemic,

$T$—time of observation ("today").

Setting $t = t_{N_F} = t_{N_{ICU}} = t_{N_{vent}}$, i.e., setting the length of the lookback period $N_B$ to be the same for all three groups of patients, $N_B \equiv N_F = N_{ICU} = N_{vent}$ reduces (19–21) to

$$H_F(t) = \begin{cases} \dfrac{\mathcal{H}_{+F}(t; N_B)}{\mathcal{I}_{+NS}(t; N_B)}, & T_0 <= t_0 < t_{N_B-1} <= T, \\[3ex] \dfrac{1}{N_B} \sum_{i=0}^{N_B-1} H_F(t_i), & t_{N_F-1} > T \end{cases} \quad , \qquad (22)$$

$$H_{ICU}(t) = \begin{cases} \dfrac{\mathcal{H}_{+ICU}(t; N_B)}{\mathcal{H}_{+F}(t; N_B)}, & T_0 <= t_0 < t_{N_B-1} <= T, \\[3ex] \dfrac{1}{N_B} \sum_{i=0}^{N_B-1} H_{ICU}(t_i), & t_{N_B-1} > T \end{cases} \quad , \qquad (23)$$

$$H_{vent}(t) = \begin{cases} \dfrac{\mathcal{H}_{+vent}(t; N_B)}{\mathcal{H}_{+ICU}(t; N_B)}, & T_0 <= t_0 < t_{N_B-1} <= T, \\[3mm] \dfrac{1}{N_B}\displaystyle\sum_{i=0}^{N_B-1} H_{vent}(t_i), & t_{N_B-1} > T \end{cases} \qquad (24)$$

In other words, hospitalization, ICU and vents rates are computed exactly as rolling $N$-day averages up until the current time, and then extrapolated as averages over the same time period going forward. It does not appear possible to define those rates smoothly since the calculation of the rate itself depends on the predicted number of affected patients which, in turn, depends on the rate creating a "circular reference".

The numbers of hospitalizations, ICU and vent placements are specific to the population served by a healthcare system and can be extrapolated to other entities only with caution. At the beginning of the pandemic, state-wide and regional data was either not available or unreliable thus necessitating an approximation using NS census and deaths. In doing so, the number of patients entering the hospital floor, ICU units and being intubated was assumed to be proportional to the observed number of cases.

In order to predict the counts (censuses) of the patient population currently hospitalized, placed in the ICU and intubated, it is necessary to model the flow of patients through each of those units. This can be done by backing out ("bootstrapping") recovery rates from the observed population dynamics as follows:

$$\mathcal{H}_F(T) = \mathcal{H}_F(T - \Delta T) + \Delta\mathcal{H}_{+F}(t) - \Delta\mathcal{H}_{-F}(t) , \qquad (25)$$

$$\mathcal{H}_{ICU}(T) = \mathcal{H}_{ICU}(T - \Delta T) + \Delta\mathcal{H}_{+ICU}(t) - \Delta\mathcal{H}_{-ICU}(t) , \qquad (26)$$

$$\mathcal{H}_{vent}(T) = \mathcal{H}_{vent}(T - \Delta T) + \Delta\mathcal{H}_{+vent}(t) - \Delta\mathcal{H}_{-vent}(t) , \qquad (27)$$

$$\Delta\mathcal{H}_{+F}(t) = H_F(t)\Delta\mathcal{I}_{+NS}(t) , \qquad (28)$$

$$\Delta\mathcal{H}_{+ICU}(t) = H_{ICU}(t)\Delta\mathcal{H}_{+F}(t) , \qquad (29)$$

$$\Delta\mathcal{H}_{+vent}(t) = H_{vent}(t)\Delta\mathcal{H}_{+ICU}(t) , \qquad (30)$$

$$\Delta\mathcal{H}_{-F}(t) = (\mu_F(t) - 1)\mathcal{H}_{-F}(t - \Delta t) + \mathcal{H}_{-F}(t) , \qquad (31)$$

$$\Delta\mathcal{H}_{-ICU}(t) = (\mu_{ICU}(t) - 1)\mathcal{H}_{-ICU}(t - \Delta t) + \mathcal{H}_{-ICU}(t) , \qquad (32)$$

$$\Delta\mathcal{H}_{-vent}(t) = (\mu_{vent}(t) - 1)\mathcal{H}_{-vent}(t - \Delta t) + \mathcal{H}_{-vent}(t) , \qquad (33)$$

where

$\mathcal{H}_F(T)$—floor census at time $t$,
$\mathcal{H}_{ICU}(t)$—ICU census at time $t$,
$\mathcal{H}_{vent}(t)$—number of ventilated patients at time $t$.
$H_F(t)$—hospitalization rate at time $t$,
$H_{ICU}(t)$—ICU placement rate at time $t$,
$H_{vent}(t)$—intubation rate at time $t$,

$\Delta\mathcal{H}_{-F}(t)$—number of patients removed (due to discharge, placement in the ICU or death) from the floor at time $t$,

$\Delta\mathcal{H}_{-ICU}(t)$—number of patients removed (due to return to the general floor population, intubation or death) from the ICU at time $t$,

$\Delta\mathcal{H}_{-vent}(t)$—number of extubations (due to extubation or death) at time $t$,

$N_F$—length of lookback period for floor patients (at the time of this writing, 7 days),

$N_{ICU}$—length of lookback period for ICU patients (7 days),

$N_{vent}$—length of lookback period for intubated patients (14 days),

$\mu_F(t)$—observed floor removal rate at time $t$,

$\mu_{ICU}(t)$—observed ICU placement rate at time $t$,

$\mu_{vent}(t)$—observed extubation rate at time $t$.

Eqs (31)–(33) can be used to determine the values of $\mu_F$, $\mu_{ICU}$ and $\mu_{vent}$ for $t \leq T$. For $t > T$, moving average extrapolations are used (cf. (22–24)):

$$\mu_F(t) = \frac{1}{N_F}\sum_{i=0}^{N_F-1}\mu_F(t_i) \, , \tag{34}$$

$$\mu_{ICU}(t) = \frac{1}{N_{ICU}}\sum_{i=0}^{N_{ICU}-1}\mu_{ICU}(t_i) \, , \tag{35}$$

$$\mu_{vent}(t) = \frac{1}{N_{vent}}\sum_{i=0}^{N_{vent}-1}\mu_{vent}(t_i) \, , \tag{36}$$

$$t \equiv t_{N_F} \equiv t_{N_{ICU}} \equiv t_{N_{vent}} \, , t_i = t_{i-1} + \Delta t \, .$$

Following the patient flow assumptions reflected in Fig 1, mortality rate is computed as

$$M(t) = \frac{\sum_{i=0}^{N}\mathcal{M}(t_i)}{\sum_{i=0}^{N}\mathcal{H}_{ICU}(t_i)} \, , \tag{37}$$

where

$M(t)$—mortality rate at time $t$,

$\mathcal{M}(t)$—cumulative number of deaths at time $t$,

$t \equiv t_N$—current time.

## Projecting vaccination rates

Vaccination rates in the CCL area at the time of this writing followed a quartic trajectory with remarkable accuracy, as shown in Fig 3.

Empirically, the shape of the quartic parabola is determined using the usual least squares best fit to be

$$\begin{aligned}\mathcal{V}_{CCL}^{(4)}(t) = {}&-5.667435 \times 10^{-3} \, t^4 + 7.590466 \, t^3 - 3.619223 \times 10^3 \, t^2 \\ &+7.302425 \times 10^5 \, t - 5.232056 \times 10^7 \, .\end{aligned} \tag{38}$$

The number of significant digits in 38 is extended for consistency.

Imposing the upper limit of 100% of the population and requiring that the number of vaccinated individuals be monotonically nondecreasing, we obtain (based on the official CCL vaccination data from Jan. 3, 2021 to May 30, 2021)

$$\mathcal{V}_{CCL}(t) = \min[N_{CCL}, \max(\mathcal{V}_{CCL}^{(4)}(t), \mathcal{V}_{CCL}(t - \Delta t))] \ . \tag{39}$$

The number of fully vaccinated NS patients is then obtained from (18) as

$$\mathcal{V}_{NS}(t) = \frac{N}{N_{CCL}}(t)\mathcal{V}_{CCL}(t) \ . \tag{40}$$

## Results and discussion

### Worked example

In the example below, we assume that CCL population is 5,846,768, relevant NS patient population estimated by (18) is 392,000 and infection transmission period $1/\gamma = 15$. We set the pandemic start date to March 10, 2020.

The date for this example was arbitrarily chosen from past history with the requirements that the number of new cases, patient admissions and censuses on the floor, in the ICU and attached to a ventilator be reasonable large to avoid instability and that the trajectory of the pandemic be fairly well established.

NS case and admission data as of Feb. 26, 2021 is presented in Table 1.

The last row of the table is incomplete because, while reliable case data is generally available up to and including the next-but-last day, case and hospitalization data is relatively reliable for the preceding day.

Calculations for the Runge-Kutta implementation of the model for CCL as of Feb. 26, 2021 are presented in Table 2, and for NS they are displayed in Table 3.

With the line corresponding to the date of 2/24/2021 as an example, the algorithm proceeds as follows:

1. Using Excel Solver's unconstrained GRG (generalized reduced gradient) nonlinear optimization with centered difference approximation and automatic scaling with constraint precision = convergence tolerance = $10^{-3}$ (or any other optimization routine), find the value of $\beta$ on the preceding day that minimizes the square of the residual between the predicted and observed number of cases at time $t$:

$$\beta(t_{i-1}) = argmin(\hat{\mathcal{I}}\ (t_0, t_i) - \mathcal{I}(t_0, t_i))^2 \ , \tag{41}$$

$$\begin{aligned}
\hat{\mathcal{I}}(t_0, t_i) \quad &= \hat{\mathcal{I}}(t_0, t_{i-1}) + [(\mathcal{S}(t_i) - \mathcal{S}(t_{i-1}) \\
&\quad - (\mathcal{V}(t_{i-1}) - \mathcal{V}(t_{i-2}))]N(t_i) \ , i = \overline{2, L}, t_L \equiv t \ ,
\end{aligned} \tag{42}$$

where
$\hat{\mathcal{I}}(t_0, t_i)$—*predicted* cumulative number of identified positive cases from the beginning of the pandemic $t_0$ to time $t_i$,
$\mathcal{I}(t_0, t_i)$—*actual* cumulative number of identified positive case from the beginning of the pandemic $t_0$ to time $t_i$,
$N(t)$—NS population at time $t_i$ (i.e., the asymptotic limit of (18)).
$t_i$—current time at the $i$-th time step.
The value of $\beta$ at which this minimum is achieved corresponds to the preceding time step, 2/23/2021, and is equal to 0.043 (rRounded to two significant digits)

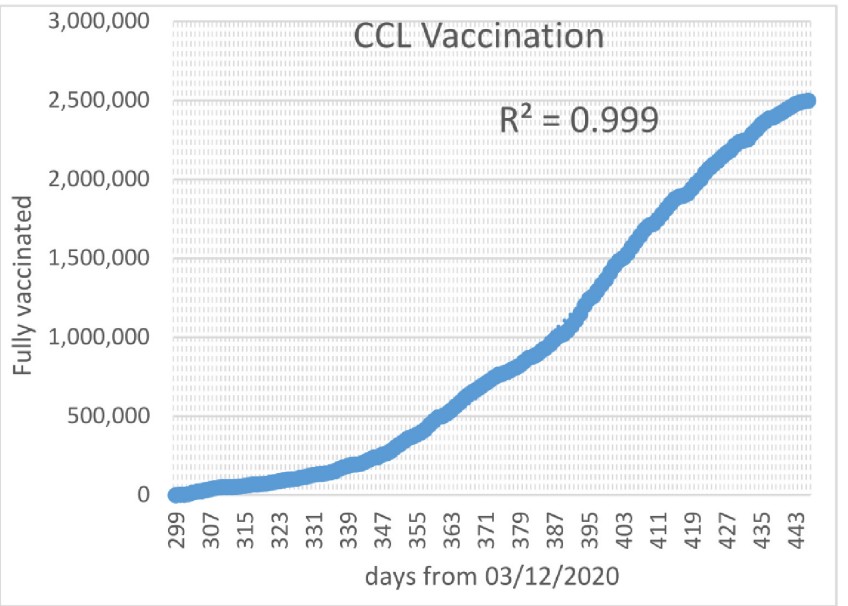

**Fig 3. CCL vaccination rates, Jan. 3.–May 30, 2021.**

2. Compute the transmission rate

$$R_0(t_i) = \frac{\beta}{\gamma} \ , i = \overline{2,L} , t_L \equiv t \ , \tag{43}$$

for future reference ($\frac{1}{\gamma}$ = 15 days, $R_0$ = 0.043 × 15 = 0.65).

3. Compute the NS susceptible rate (cf. 12)

$$S(t_i) = S(t_{i-1}) + k(t_{i-1})dt \ , \tag{44}$$

$$k(t_{i-1}) = \frac{h}{6}[k_1(t_{i-1}) + 2k_2(t_{i-1}) + 2k_3(t_{i-1}) + k_4(t_{i-1})] \ , \tag{45}$$

$$h = t_i - t_{i-1} \ , i = \overline{2,L} , t_L \equiv t \ .$$

The above yields $S(t_i)$ = 0.8604 − 6.35 × 10$^{-4}$ × 1 = 0.8598.

4. Compute the NS vaccination rate from (39 and 40) and Table 3 (significant digits added for consistency)

$$\mathcal{V}_{\mathcal{CCL}}(t_i) = \min\{5,846,768; \quad -5.667435 \times 10^{-3} \times 351^4 + 7.590466 \times 351^3$$
$$-3.619223 \times 10^3 \times 351^2 + 7.302425 \times 10^5 \times 351$$
$$-5.232056 \times 10^7\} = 318,279 \ ,$$

$$V_{CCL}(t_i) \quad = V_{NS}(t_i) = \frac{318,279}{5,846,768} = 0.054437 \ ,$$

$$\mathcal{V}_{NS}(t) \quad = 0.054437 \times 392,000 = 21,339 \ .$$

**Table 1. NS COVID-19 case and admission data, Feb. 1–24, 2021.**

| Date | Chicago+Cook+Lake | | NorthShore | | | | | | | | | | | | | | | |
|---|---|---|---|---|---|---|---|---|---|---|---|---|---|---|---|---|---|---|
| | CCL Cases | CCL Vac | NS act cases | NS Vac | NS act. Inp. | NS act. Inp. Census | NS act. ICU | Ns act. add'l ICU | NS act ICU only | GAU | NS act. ICU census | Ns act. ICU rate | NS act. Vent | Ns act. add'l vent | NS act. vent census | NS act. vent rate | NS act. Mort. | NS act. Mort. Rate |
| 2/1/2021 | 510592 | 106260 | 35701 | 7430 | 2,899 | 61 | 546 | 0 | 8 | 0 | 8 | 0.12 | 270 | 0 | 5 | 0.38 | 496 | 0.908 |
| 2/2/2021 | 511409 | 110902 | 35748 | 7752 | 2,910 | 58 | 548 | 2 | 9 | 0 | 9 | 0.14 | 272 | 2 | 6 | 0.44 | 498 | 0.909 |
| 2/3/2021 | 512613 | 117264 | 35808 | 8191 | 2,920 | 54 | 549 | 1 | 9 | 0 | 9 | 0.13 | 272 | 0 | 5 | 0.41 | 500 | 0.911 |
| 2/4/2021 | 513929 | 122009 | 35861 | 8514 | 2,926 | 51 | 553 | 4 | 11 | 0 | 11 | 0.21 | 273 | 1 | 5 | 0.3 | 502 | 0.908 |
| 2/5/2021 | 515469 | 126389 | 35901 | 8803 | 2,931 | 49 | 554 | 1 | 10 | 0 | 10 | 0.18 | 276 | 3 | 7 | 0.4 | 502 | 0.906 |
| 2/6/2021 | 516772 | 127857 | 35942 | 8893 | 2,940 | 51 | 555 | 1 | 10 | 0 | 10 | 0.18 | 277 | 1 | 7 | 0.47 | 503 | 0.906 |
| 2/7/2021 | 517677 | 132925 | 35967 | 9235 | 2,943 | 49 | 555 | 0 | 9 | 0 | 9 | 0.16 | 277 | 0 | 6 | 0.44 | 504 | 0.908 |
| 2/8/2021 | 518344 | 138618 | 36022 | 9633 | 2,953 | 52 | 556 | 1 | 9 | 0 | 9 | 0.19 | 277 | 0 | 5 | 0.47 | 505 | 0.908 |
| 2/9/2021 | 519080 | 146292 | 36074 | 10167 | 2,960 | 49 | 557 | 1 | 9 | 0 | 9 | 0.18 | 277 | 0 | 4 | 0.41 | 506 | 0.908 |
| 2/10/2021 | 520153 | 159414 | 36124 | 11071 | 2,967 | 45 | 558 | 1 | 8 | 0 | 8 | 0.19 | 278 | 1 | 3 | 0.47 | 508 | 0.910 |
| 2/11/2021 | 521218 | 171525 | 36155 | 11898 | 2,976 | 45 | 559 | 1 | 7 | 0 | 7 | 0.12 | 278 | 0 | 3 | 0.44 | 509 | 0.911 |
| 2/12/2021 | 522383 | 179935 | 36190 | 12466 | 2,982 | 43 | 560 | 1 | 7 | 0 | 7 | 0.12 | 278 | 0 | 2 | 0.5 | 510 | 0.911 |
| 2/13/2021 | 523254 | 184662 | 36228 | 12785 | 2,987 | 43 | 560 | 0 | 7 | 0 | 7 | 0.11 | 279 | 1 | 3 | 0.6 | 511 | 0.913 |
| 2/14/2021 | 523920 | 187490 | 36256 | 12975 | 2,994 | 45 | 560 | 0 | 6 | 0 | 6 | 0.1 | 279 | 0 | 1 | 0.64 | 513 | 0.916 |
| 2/15/2021 | 524508 | 192601 | 36288 | 13325 | 2,999 | 45 | 562 | 2 | 8 | 0 | 8 | 0.13 | 280 | 1 | 2 | 0.63 | 513 | 0.913 |
| 2/16/2021 | 525044 | 203676 | 36319 | 14089 | 3,003 | 43 | 562 | 0 | 8 | 0 | 8 | 0.12 | 280 | 0 | 2 | 0.57 | 514 | 0.915 |
| 2/17/2021 | 525722 | 217944 | 36358 | 15073 | 3,008 | 37 | 562 | 0 | 8 | 0 | 8 | 0.1 | 280 | 0 | 2 | 0.62 | 515 | 0.916 |
| 2/18/2021 | 526379 | 231601 | 36391 | 16012 | 3,016 | 37 | 564 | 2 | 9 | 0 | 9 | 0.13 | 280 | 0 | 2 | 0.64 | 516 | 0.915 |
| 2/19/2021 | 527198 | 234098 | 36425 | 16174 | 3,018 | 31 | 565 | 1 | 8 | 0 | 8 | 0.14 | 281 | 1 | 3 | 0.45 | 516 | 0.913 |
| 2/20/2021 | 527997 | 248822 | 36437 | 17171 | 3,020 | 27 | 565 | 0 | 8 | 0 | 8 | 0.15 | 281 | 0 | 3 | 0.4 | 518 | 0.917 |
| 2/21/2021 | 528646 | 257625 | 36456 | 17766 | 3,025 | 30 | 565 | 0 | 5 | 0 | 5 | 0.16 | 281 | 0 | 2 | 0.4 | 519 | 0.919 |
| 2/22/2021 | 529161 | 271716 | 36494 | 18739 | 3,032 | 34 | 567 | 2 | 7 | 0 | 7 | 0.15 | 284 | 3 | 5 | 0.64 | 519 | 0.915 |
| 2/23/2021 | 529797 | 293847 | 36520 | 20255 | 3,035 | 34 | 567 | 0 | 7 | 0 | 7 | 0.16 | 284 | 0 | 5 | 0.7 | 519 | 0.915 |
| 2/24/2021 | 530681 | | | | | 30 | | 0 | 6 | 0 | | | | 0 | 4 | | 519 | |

**Table 2. Runge-Kutta implementation for the CCL COVID-19 model, Feb. 1–24, 2021.**

| Date | Beta | R0 | CCL Suscept rate | CCL Vac rate | Inf rate | k1 | l1 | k2 | l2 | k3 | l3 | k4 | l4 | k | l |
|---|---|---|---|---|---|---|---|---|---|---|---|---|---|---|---|
| 2/1/2021 | 0.031 | 0.472 | 0.8958 | 0.018 | 0.0051 | -1.05E-03 | -1.02E-04 | -1.05E-03 | -1.02E-04 | -1.05E-03 | -1.02E-04 | -1.04E-03 | -1.01E-04 | -1.05E-03 | -1.02E-04 |
| 2/2/2021 | 0.048 | 0.718 | 0.8947 | 0.019 | 0.0050 | -1.01E-03 | -1.18E-04 | -1.00E-03 | -1.17E-04 | -1.00E-03 | -1.17E-04 | -1.00E-03 | -1.17E-04 | -1.00E-03 | -1.17E-04 |
| 2/3/2021 | 0.054 | 0.803 | 0.8937 | 0.020 | 0.0048 | -1.32E-03 | -9.11E-05 | -1.32E-03 | -9.05E-05 | -1.32E-03 | -9.05E-05 | -1.32E-03 | -9.05E-05 | -1.32E-03 | -9.06E-05 |
| 2/4/2021 | 0.064 | 0.955 | 0.8924 | 0.021 | 0.0048 | -1.08E-03 | -4.69E-05 | -1.08E-03 | -4.69E-05 | -1.08E-03 | -4.69E-05 | -1.08E-03 | -4.69E-05 | -1.08E-03 | -4.69E-05 |
| 2/5/2021 | 0.055 | 0.820 | 0.8913 | 0.022 | 0.0047 | -9.78E-04 | -8.44E-05 | -9.76E-04 | -8.38E-05 | -9.76E-04 | -8.38E-05 | -9.76E-04 | -8.38E-05 | -9.77E-04 | -8.39E-05 |
| 2/6/2021 | 0.039 | 0.584 | 0.8903 | 0.022 | 0.0046 | -4.11E-04 | -1.48E-04 | -4.09E-04 | -1.45E-04 | -4.09E-04 | -1.45E-04 | -4.09E-04 | -1.45E-04 | -4.09E-04 | -1.46E-04 |
| 2/7/2021 | 0.030 | 0.447 | 0.8899 | 0.023 | 0.0045 | -9.85E-04 | -1.80E-04 | -9.83E-04 | -1.76E-04 | -9.83E-04 | -1.76E-04 | -9.83E-04 | -1.76E-04 | -9.83E-04 | -1.77E-04 |
| 2/8/2021 | 0.034 | 0.513 | 0.8889 | 0.024 | 0.0043 | -1.10E-03 | -1.56E-04 | -1.10E-03 | -1.53E-04 | -1.10E-03 | -1.53E-04 | -1.10E-03 | -1.53E-04 | -1.10E-03 | -1.54E-04 |
| 2/9/2021 | 0.051 | 0.771 | 0.8878 | 0.025 | 0.0041 | -1.50E-03 | -8.72E-05 | -1.50E-03 | -8.65E-05 | -1.50E-03 | -8.65E-05 | -1.50E-03 | -8.65E-05 | -1.50E-03 | -8.66E-05 |
| 2/10/2021 | 0.052 | 0.783 | 0.8863 | 0.027 | 0.0041 | -2.43E-03 | -8.28E-05 | -2.43E-03 | -8.23E-05 | -2.43E-03 | -8.23E-05 | -2.43E-03 | -8.23E-05 | -2.43E-03 | -8.24E-05 |
| 2/11/2021 | 0.058 | 0.874 | 0.8839 | 0.029 | 0.0040 | -2.28E-03 | -6.03E-05 | -2.27E-03 | -6.01E-05 | -2.27E-03 | -6.01E-05 | -2.27E-03 | -6.01E-05 | -2.27E-03 | -6.01E-05 |
| 2/12/2021 | 0.045 | 0.669 | 0.8816 | 0.031 | 0.0039 | -1.59E-03 | -1.07E-04 | -1.59E-03 | -1.06E-04 | -1.59E-03 | -1.06E-04 | -1.59E-03 | -1.06E-04 | -1.59E-03 | -1.06E-04 |
| 2/13/2021 | 0.035 | 0.529 | 0.8800 | 0.032 | 0.0038 | -9.27E-04 | -1.36E-04 | -9.24E-04 | -1.33E-04 | -9.24E-04 | -1.33E-04 | -9.24E-04 | -1.33E-04 | -9.25E-04 | -1.34E-04 |
| 2/14/2021 | 0.032 | 0.485 | 0.8791 | 0.032 | 0.0037 | -5.88E-04 | -1.41E-04 | -5.86E-04 | -1.38E-04 | -5.86E-04 | -1.38E-04 | -5.86E-04 | -1.38E-04 | -5.86E-04 | -1.38E-04 |
| 2/15/2021 | 0.031 | 0.460 | 0.8785 | 0.033 | 0.0035 | -9.69E-04 | -1.41E-04 | -9.67E-04 | -1.38E-04 | -9.68E-04 | -1.38E-04 | -9.68E-04 | -1.38E-04 | -9.68E-04 | -1.38E-04 |
| 2/16/2021 | 0.040 | 0.604 | 0.8776 | 0.035 | 0.0034 | -2.01E-03 | -1.07E-04 | -2.01E-03 | -1.05E-04 | -2.01E-03 | -1.05E-04 | -2.01E-03 | -1.05E-04 | -2.01E-03 | -1.05E-04 |
| 2/17/2021 | 0.040 | 0.605 | 0.8755 | 0.037 | 0.0033 | -2.56E-03 | -1.03E-04 | -2.55E-03 | -1.02E-04 | -2.55E-03 | -1.02E-04 | -2.55E-03 | -1.02E-04 | -2.56E-03 | -1.02E-04 |
| 2/18/2021 | 0.052 | 0.777 | 0.8730 | 0.040 | 0.0032 | -2.48E-03 | -6.84E-05 | -2.48E-03 | -6.79E-05 | -2.48E-03 | -6.79E-05 | -2.48E-03 | -6.79E-05 | -2.48E-03 | -6.80E-05 |
| 2/19/2021 | 0.052 | 0.776 | 0.8705 | 0.040 | 0.0031 | -5.68E-04 | -6.76E-05 | -5.66E-04 | -6.69E-05 | -5.66E-04 | -6.69E-05 | -5.66E-04 | -6.69E-05 | -5.66E-04 | -6.70E-05 |
| 2/20/2021 | 0.043 | 0.648 | 0.8699 | 0.043 | 0.0031 | -2.63E-03 | -8.89E-05 | -2.63E-03 | -8.78E-05 | -2.63E-03 | -8.78E-05 | -2.63E-03 | -8.78E-05 | -2.63E-03 | -8.80E-05 |
| 2/21/2021 | 0.035 | 0.532 | 0.8673 | 0.044 | 0.0030 | -1.60E-03 | -1.07E-04 | -1.60E-03 | -1.05E-04 | -1.60E-03 | -1.05E-04 | -1.60E-03 | -1.05E-04 | -1.60E-03 | -1.05E-04 |
| 2/22/2021 | 0.045 | 0.680 | 0.8657 | 0.046 | 0.0029 | -2.52E-03 | -7.85E-05 | -2.52E-03 | -7.76E-05 | -2.52E-03 | -7.76E-05 | -2.52E-03 | -7.76E-05 | -2.52E-03 | -7.77E-05 |
| 2/23/2021 | 0.064 | 0.967 | 0.8629 | 0.047 | 0.0027 | -7.26e-04 | -3.02E-05 | -7.25E-04 | -3.01E-05 | -7.25E-04 | -3.01E-05 | -7.24E-04 | -3.00E-05 | -7.25E-04 | -3.01E-05 |
| 2/24/2021 | 0.048 | 0.721 | 0.8622 | 0.054 | 0.0027 | -7.50e-03 | -6.84E-05 | -7.50E-03 | -6.80E-05 | -7.50E-03 | -6.80E-05 | -7.50e-03 | -6.76E-05 | -7.50E-03 | -6.80E-05 |

5. Compute the number of patients who are no longer susceptible to the disease

$$\rho_{NS}(t_i) = (1 - S(t_i))N(t_i) = (1 - 0.8598) \times 392,000 = 54,967 \,,$$

6. Compute the infection rate

$$I(t_i) = I(t_{i-1}) + l(t_{i-1})dt \,, \tag{46}$$

$$l(t_{i-1}) = \frac{h}{6}\left[l_1(t_{i-1}) + 2l_2(t_{i-1}) + 2l_3(t_{i-1}) + l_4(t_{i-1})\right] \,, \tag{47}$$

$$h = t_i - t_{i-1} \,, i = \overline{2, L}, t_L \equiv t \,.$$

The above yields $I(t_i) = 1.63 \times 10^{-3} - 4.88 \times 10^{-5} = 1.63 \times 10^{-3}$.

7. Compute the predicted cumulative number of positive cases from (42) to arrive at

$$\mathcal{I}(t_i) = 36,501 + [(0.86041 - 0.85978) - (0.0493 - 0.0470) \times 10^{-3}]$$
$$\times 392,000 = 36,525 \,.$$

The predicted number of new infections is $36,525 - 36,501 = 24$ (In reality, this calculation is performed in reverse order).

**Table 3. Runge-Kutta implementation for the NS COVID-19 model, Feb. 1–24, 2021.**

| Date | Day | NS beta | NS R0 | Vac Rate | NS Suscept rate | Not suscept # | Inf # | Inf cumul | Inf rate | NS pred hosp rate | Cu-mul hosp # | Ho-sp Δ | NS inp disch rate | IP cen-sus | NS pred ICU rate | Cu-mul ICU # | ICU Δ | NS ICU disch rate | ICU census | NS pred vent rate | Cu-mul Vent # | Vent? |
|---|---|---|---|---|---|---|---|---|---|---|---|---|---|---|---|---|---|---|---|---|---|---|
| 2/1/21 | 321 | 0.046 | 0.70 | 0.018 | 0.8921 | 42,315 | 1,151 | 35,683 | 2.94E-03 | 0.135 | 2905 | 11 | 0.167 | 61 | 0.138 | 546 | 0 | 0.11 | 8 | 0.38 | 270 | 0 |
| 2/2/21 | 322 | 0.060 | 0.91 | 0.019 | 0.8907 | 42,854 | 1,122 | 35,730 | 2.86E-03 | 0.138 | 2916 | 11 | 0.230 | 58 | 0.140 | 548 | 2 | 0.125 | 9 | 0.44 | 272 | 2 |
| 2/3/21 | 323 | 0.054 | 0.81 | 0.020 | 0.8897 | 43,225 | 1,108 | 35,790 | 2.83E-03 | 0.143 | 2926 | 10 | 0.241 | 54 | 0.145 | 549 | 1 | 0.111 | 9 | 0.41 | 272 | 0 |
| 2/4/21 | 324 | 0.042 | 0.63 | 0.021 | 0.8885 | 43,705 | 1,088 | 35,843 | 2.77E-03 | 0.146 | 2932 | 6 | 0.167 | 51 | 0.174 | 553 | 4 | 0.222 | 11 | 0.30 | 273 | 1 |
| 2/5/21 | 325 | 0.044 | 0.67 | 0.022 | 0.8876 | 44,063 | 1,056 | 35,883 | 2.69E-03 | 0.151 | 2937 | 5 | 0.137 | 49 | 0.179 | 554 | 1 | 0.182 | 10 | 0.40 | 276 | 3 |
| 2/6/21 | 326 | 0.028 | 0.42 | 0.022 | 0.8867 | 44,398 | 1,028 | 35,924 | 2.62E-03 | 0.156 | 2946 | 9 | 0.143 | 51 | 0.167 | 555 | 1 | 0.100 | 10 | 0.47 | 277 | 1 |
| 2/7/21 | 327 | 0.063 | 0.95 | 0.023 | 0.8864 | 44,521 | 986 | 35,949 | 2.51E-03 | 0.153 | 2949 | 3 | 0.098 | 49 | 0.167 | 555 | 0 | 0.100 | 9 | 0.44 | 277 | 0 |
| 2/8/21 | 328 | 0.062 | 0.93 | 0.024 | 0.8854 | 44,916 | 975 | 36,004 | 2.49E-03 | 0.160 | 2959 | 10 | 0.143 | 52 | 0.153 | 556 | 1 | 0.111 | 9 | 0.47 | 277 | 0 |
| 2/9/21 | 329 | 0.059 | 0.89 | 0.025 | 0.8843 | 45,351 | 964 | 36,057 | 2.46E-03 | 0.159 | 2966 | 7 | 0.192 | 49 | 0.156 | 557 | 1 | 0.111 | 9 | 0.41 | 277 | 1 |
| 2/10/21 | 330 | 0.038 | 0.56 | 0.027 | 0.8829 | 45,915 | 950 | 36,107 | 2.42E-03 | 0.159 | 2973 | 7 | 0.224 | 45 | 0.159 | 558 | 1 | 0.222 | 8 | 0.47 | 278 | 1 |
| 2/11/21 | 331 | 0.044 | 0.66 | 0.029 | 0.8805 | 46,826 | 919 | 36,138 | 2.34E-03 | 0.164 | 2982 | 9 | 0.200 | 45 | 0.168 | 559 | 1 | 0.250 | 7 | 0.44 | 278 | 0 |
| 2/12/21 | 332 | 0.049 | 0.74 | 0.031 | 0.8784 | 47,673 | 893 | 36,173 | 2.28E-03 | 0.175 | 2989 | 7 | 0.178 | 44 | 0.150 | 560 | 1 | 0.143 | 7 | 0.50 | 278 | 0 |
| 2/13/21 | 333 | 0.037 | 0.56 | 0.032 | 0.8768 | 48,275 | 872 | 36,211 | 2.23E-03 | 0.170 | 2994 | 5 | 0.114 | 44 | 0.144 | 560 | 0 | 0.000 | 7 | 0.60 | 279 | 1 |
| 2/14/21 | 334 | 0.044 | 0.66 | 0.032 | 0.8760 | 48,620 | 843 | 36,239 | 2.15E-03 | 0.177 | 3001 | 7 | 0.136 | 45 | 0.131 | 560 | 0 | 0.143 | 6 | 0.64 | 279 | 0 |
| 2/15/21 | 335 | 0.042 | 0.64 | 0.033 | 0.8754 | 48,841 | 820 | 36,271 | 2.09E-03 | 0.172 | 3006 | 5 | 0.111 | 45 | 0.158 | 562 | 2 | 0.000 | 8 | 0.63 | 280 | 1 |
| 2/16/21 | 336 | 0.057 | 0.85 | 0.035 | 0.8745 | 49,214 | 796 | 36,301 | 2.03E-03 | 0.165 | 3010 | 4 | 0.133 | 43 | 0.149 | 562 | 0 | 0.000 | 8 | 0.57 | 280 | 0 |
| 2/17/21 | 337 | 0.049 | 0.74 | 0.037 | 0.8725 | 49,996 | 782 | 36,340 | 2.00E-03 | 0.162 | 3015 | 5 | 0.256 | 37 | 0.146 | 562 | 0 | 0.000 | 8 | 0.62 | 280 | 0 |
| 2/18/21 | 338 | 0.050 | 0.75 | 0.040 | 0.8699 | 50,985 | 764 | 36,373 | 1.95E-03 | 0.172 | 3023 | 8 | 0.216 | 37 | 0.121 | 564 | 2 | 0.125 | 9 | 0.64 | 280 | 0 |
| 2/19/21 | 339 | 0.019 | 0.29 | 0.040 | 0.8675 | 51,934 | 746 | 36,406 | 1.90E-03 | 0.168 | 3025 | 2 | 0.216 | 31 | 0.125 | 565 | 1 | 0.222 | 8 | 0.45 | 281 | 1 |
| 2/20/21 | 340 | 0.032 | 0.47 | 0.043 | 0.8671 | 52,113 | 710 | 36,418 | 1.81E-03 | 0.164 | 3027 | 2 | 0.194 | 27 | 0.123 | 565 | 0 | 0.000 | 8 | 0.40 | 281 | 0 |
| 2/21/21 | 341 | 0.065 | 0.97 | 0.044 | 0.8645 | 53,119 | 682 | 36,437 | 1.74E-03 | 0.170 | 3032 | 5 | 0.074 | 30 | 0.12 | 565 | 0 | 0.375 | 5 | 0.40 | 281 | 0 |
| 2/22/21 | 342 | 0.045 | 0.68 | 0.046 | 0.8629 | 53,748 | 675 | 36,475 | 1.72E-03 | 0.172 | 3040 | 8 | 0.100 | 35 | 0.136 | 567 | 2 | 0.000 | 7 | 0.64 | 284 | 3 |
| 2/23/21 | 343 | 0.043 | 0.65 | 0.047 | 0.8604 | 54,718 | 657 | 36,501 | 1.68E-03 | 0.173 | 3043 | 3 | 0.086 | 35 | 0.13 | 567 | 0 | 0.000 | 7 | 0.70 | 284 | 0 |
| 2/24/21 | 344 | 0.061 | 0.91 | 0.054 | 0.8598 | 54,967 | 638 | 36,525 | 1.63E-03 | 0.169 | 3044 | 1 | 0.170 | 30 | 0.140 | 568 | 1 | 0.106 | 7 | 0.55 | 284 | 0 |

| Date | NS vent disch rate | Vent cen-sus | NS pr-ed mo-rt rate | NS pr-ed mo-rt | NS new inf | k1 | l1 | k2 | l2 | k3 | l3 | k4 | l4 | k | l |
|---|---|---|---|---|---|---|---|---|---|---|---|---|---|---|---|
| 2/1/21 | 0.00 | 5 | 0.910 | 497 | 50 | -1.38E-03 | -7.42E-05 | -1.38E-03 | -7.34E-05 | -1.38E-03 | -7.34E-05 | -1.38E-03 | -7.25E-05 | -1.38E-05 | -7.34E-05 |
| 2/2/21 | 0.20 | 6 | 0.911 | 499 | 47 | -9.48E-04 | -3.67E-05 | -9.47E-04 | -3.65E-05 | -9.47E-04 | -3.65E-05 | -9.46E-04 | -3.64E-05 | -9.47E-05 | -3.65E-05 |
| 2/3/21 | 0.17 | 5 | 0.913 | 501 | 60 | -1.22E-03 | -5.19E-05 | -1.22E-03 | -5.15E-05 | -1.22E-03 | -5.15E-05 | -1.22E-03 | -5.11E-05 | -1.22E-05 | -5.15E-05 |
| 2/4/21 | 0.20 | 5 | 0.910 | 503 | 53 | -9.15E-04 | -8.14E-05 | -9.14E-04 | -8.02E-05 | -9.14E-04 | -8.02E-05 | -9.12E-04 | -7.91E-05 | -9.14E-05 | -8.02E-05 |
| 2/5/21 | 0.20 | 7 | 0.908 | 503 | 40 | -8.55E-04 | -7.35E-05 | -8.54E-04 | -7.26E-05 | -8.54E-04 | -7.26E-05 | -8.52E-04 | -7.17E-05 | -8.54E-05 | -7.26E-05 |
| 2/6/21 | 0.14 | 7 | 0.908 | 504 | 41 | -3.16E-04 | -1.10E-04 | -3.15E-04 | -1.07E-04 | -3.15E-04 | -1.07E-04 | -3.14E-04 | -1.05E-04 | -3.15E-04 | -1.07E-04 |
| 2/7/21 | 0.14 | 6 | 0.910 | 505 | 25 | -1.01E-03 | -2.65E-05 | -1.01E-03 | -2.64E-05 | -1.01E-03 | -2.64E-05 | -1.01E-03 | -2.64E-05 | -1.01E-03 | -2.64E-05 |
| 2/8/21 | 0.17 | 5 | 0.910 | 506 | 55 | -1.11E-03 | -2.98E-05 | -1.11E-03 | -2.97E-05 | -1.11E-03 | -2.97E-05 | -1.11E-03 | -2.96E-05 | -1.11E-03 | -2.97E-05 |
| 2/9/21 | 0.20 | 4 | 0.910 | 507 | 53 | -1.44E-03 | -3.53E-05 | -1.44E-03 | -3.52E-05 | -1.44E-03 | -3.52E-05 | -1.44E-03 | -3.50E-05 | -1.44E-03 | -3.52E-05 |
| 2/10/21 | 0.50 | 3 | 0.912 | 509 | 50 | -2.32E-03 | -8.10E-05 | -2.32E-03 | -7.98E-05 | -2.32E-03 | -7.98E-05 | -2.32E-03 | -7.86E-05 | -2.32E-05 | -7.98E-05 |
| 2/11/21 | 0.00 | 3 | 0.912 | 510 | 31 | -2.16E-03 | -6.56E-05 | -2.16E-03 | -6.48E-05 | -2.16E-03 | -6.48E-05 | -2.16E-03 | -6.40E-05 | -2.16E-05 | -6.48E-05 |
| 2/12/21 | 0.33 | 2 | 0.913 | 511 | 35 | -1.54E-03 | -5.37E-05 | -1.54E-03 | -5.32E-05 | -1.54E-03 | -5.32E-05 | -1.53E-03 | -5.26E-05 | -1.54E-05 | -5.32E-05 |
| 2/13/21 | 0.00 | 3 | 0.914 | 512 | 38 | -8.81E-04 | -7.57E-05 | -8.80E-04 | -7.44E-05 | -8.80E-04 | -7.44E-05 | -8.79E-04 | -7.32E-05 | -8.80E-04 | -7.44E-05 |
| 2/14/21 | 0.67 | 1 | 0.918 | 514 | 28 | -5.67E-04 | -6.06E-05 | -5.65E-04 | -5.97E-05 | -5.65E-04 | -5.98E-05 | -5.64E-04 | -5.89E-05 | -5.65E-04 | -5.98E-05 |
| 2/15/21 | 0.00 | 2 | 0.915 | 514 | 32 | -9.52E-04 | -6.17E-05 | -9.51E-04 | -6.08E-05 | -9.51E-04 | -6.08E-05 | -9.50E-04 | -6.00E-05 | -9.51E-04 | -6.08E-05 |
| 2/16/21 | 0.00 | 2 | 0.916 | 515 | 30 | -1.99E-03 | -3.49E-05 | -1.99E-03 | -3.47E-05 | -1.99E-03 | -3.47E-05 | -1.99E-03 | -3.45E-05 | -1.99E-03 | -3.47E-05 |
| 2/17/21 | 0.00 | 2 | 0.918 | 516 | 39 | -2.53E-03 | -4.77E-05 | -2.52E-03 | -4.73E-05 | -2.52E-03 | -4.73E-05 | -2.52E-03 | -4.68E-05 | -2.52E-03 | -4.73E-05 |
| 2/18/21 | 0.00 | 2 | 0.917 | 517 | 33 | -2.42E-03 | -4.46E-05 | -2.42E-03 | -4.42E-05 | -2.42E-03 | -4.42E-05 | -2.42E-03 | -4.38E-05 | -2.42E-03 | -4.42E-05 |
| 2/19/21 | 0.00 | 3 | 0.915 | 517 | 34 | -4.58E-04 | -9.55E-05 | -4.58E-04 | -9.32E-05 | -4.58E-04 | -9.32E-05 | -4.57E-04 | -9.09E-05 | -4.58E-04 | -9.32E-05 |
| 2/20/21 | 0.00 | 3 | 0.919 | 519 | 12 | -2.57E-03 | -7.12E-05 | -2.57E-03 | -6.99E-05 | -2.57E-03 | -6.99E-05 | -2.57E-03 | -6.86E-05 | -2.57E-05 | -6.99E-05 |
| 2/21/21 | 0.33 | 2 | 0.920 | 520 | 19 | -1.60E-03 | -1.85E-05 | -1.60E-03 | -1.85E-05 | -1.60E-03 | -1.85E-05 | -1.60E-03 | -1.85E-05 | -1.60E-03 | -1.85E-05 |
| 2/22/21 | 0.00 | 5 | 0.917 | 520 | 38 | -2.48E-03 | -4.75E-05 | -2.48E-03 | -4.69E-05 | -2.48E-03 | -4.69E-05 | -2.48E-03 | -4.64E-05 | -2.48E-03 | -4.69E-05 |
| 2/23/21 | 0.00 | 5 | 0.917 | 520 | 26 | -6.35E-04 | -4.95E-05 | -6.35E-04 | -4.88E-05 | -6.35E-04 | -4.88E-05 | -6.34E-04 | -4.81E-05 | -6.35E-05 | -4.88E-05 |
| 2/24/21 | 0.13 | 4 | 0.917 | 520 | 24 | -7.48E-03 | -2.35E-05 | -7.47E-03 | -2.37E-05 | -7.47E-03 | -2.37E-05 | -7.47E-03 | -2.39E-05 | -7.47E-03 | -2.37E-05 |

8. Compute the predicted hospitalization rate from (22)

$$H_F(t_i) \quad = 0.1687 \ .$$

9. Compute cumulative hospitalizations

$$
\begin{aligned}
\mathcal{H}_F(t_0; t_i) \quad &= \max(\mathcal{H}_F(t_0; t_i), \mathcal{H}_F(t_0; t_{i-N_F}) + H_F(t_i)(\Delta \mathcal{I}(t_0; t_i) \\
&\quad - \Delta \mathcal{I}(t_0; t_i - N_F))) = \max(3,043, 2,973 + 0.1687 \\
&\quad \times (36,525 - 36,107)) = 3,044 \ .
\end{aligned}
$$

and $\Delta \mathcal{H}_{+F}(t_i) = 3,044 - 3,043 = 1$ new patient.

10. Compute the predicted floor removal rate from (34): $\mu_F(t_i) = 0.170$.

11. Compute the predicted floor census from (25)

$$\mathcal{H}_F(t_i) = 4 + 35 \times (1 - 0.170) = 33 \ .$$

12. Repeat steps 8–11 for ICU and vented patients to obtain from (26–29)

$$
\begin{aligned}
H_{ICU}(t_i) \quad &= 0.140; , \\
\mathcal{H}_{ICU}(t_0; t_i) \quad &= \max(\mathcal{H}_{ICU}(t_0; t_i), \mathcal{H}_{ICU}(t_0; t_{i-N_F}) \\
&\quad + H_{ICU}(t_i)(\mathcal{H}_{\mathcal{F}}(t_0; t_i) - \mathcal{H}_{\mathcal{F}}(t_0; t_i - N_F)) \\
&= \max(567, 558 + 0.140 \times (3,044 - 2,973)) = 568 \ , \\
\Delta \mathcal{H}_{+ICU}(t_i) \quad &= 568 - 567 = 1 \ , \\
\mu_{ICU}(t_i) \quad &= 0.106 \ , \\
\mathcal{H}_{ICU} \quad &= 1 + 7 \times (1 - 0.106) = 7 \ , \\
\mathcal{H}_{vent}(t_0; t_i) \quad &= \max(\mathcal{H}_{vent}(t_0; t_i), \mathcal{H}_{vent}(t_0; t_{i-N_F}) \\
&\quad + H_{vent}(t_i)(\mathcal{H}_{ICU}(t_0; t_i) - \mathcal{H}_{ICU}(t_0; t_i - N_F)) \\
&= \max(284, 278 + 0.55 \times (567 - 558)) = 284 \ , \\
\Delta \mathcal{H}_{+vent}(t_i) \quad &= 284 - 284 = 0 \ , \\
\mu_{vent}(t_i) \quad &= 0.130 \ , \\
\mathcal{H}_{vent} \quad &= 0 + 5 \times (1 - 0.13) = 4 \ .
\end{aligned}
$$

13. Compute expected mortality rate and mortality from (37)

$$M(t_i) \quad = \frac{\mathcal{M}(t_i)}{\mathcal{H}_{ICU}(t_0; t_i)} = \frac{520}{567} = 0.917 \ .$$

## Model accuracy

An analysis of the accuracy of predictions was performed for the period starting on Mar. 10, 2020 and ending on May 24, 2021. Predictions were initially generated daily, then weekly, then

**Table 4. Accuracy of the NS COVID-19 model, Mar. 10, 2020–May 24, 2021.**

| Variable | Positive cases | | | Inpatient census | | | ICU census | | | Vent census | | | Cumul. Mortality | | |
|---|---|---|---|---|---|---|---|---|---|---|---|---|---|---|---|
| | Pred. 1 day ago | Pred. 1 wk. ago | Pred. 2 wks. ago | Pred. 1 day ago | Pred. 1 wk. ago | Pred. 2 wks. ago | Pred. 1 day ago | Pred. 1 wk. ago | Pred. 2 wks. ago | Pred. 1 day ago | Pred. 1 wk. ago | Pred. 2 wks. ago | Pred. 1 day ago | Pred. 1 wk. ago | Pred. 2 wks. ago |
| % correct | 32% | 27% | 21% | 38% | 37% | 29% | 25% | 31% | 23% | 20% | 27% | 28% | 34% | 33% | 32% |
| med % err. | 0% | 0% | 1% | 4% | 6% | 18% | 13% | 11% | 20% | 18% | 18% | 24% | 1% | 1% | 2% |
| av. % err. | 2% | 5% | 16% | 10% | 15% | 33% | 21% | 26% | 40% | 30% | 39% | 47% | 2% | 3% | 3% |
| std. % err. | 8% | 29% | 139% | 14% | 22% | 95% | 35% | 55% | 106% | 47% | 67% | 89% | 3% | 3% | 3% |

twice a week, on Mondays and Thursdays with a few exceptions around statutory holiday. Forecasts produced between the periods of calculation were classified as "one-day-ahead" (even though they may have been issued 1 to 6 days in advance); one- and two-week-ahead predictions were also considered.

Two sets of predictions were issued on each occasion: one based on the NS data, the other extrapolated from the CCL data adjusting for the then-current share that NS population represented in the CCL pool according to (17 and 18). The synthesis of two disparate sources required a different metric than, e.g., weighted interval score, employed for this purpose in [31, 32]. For practical purposes, we adopted a simplified approach described below (in general, our conclusions about the accuracy of the developed model, although arrived at through different means, are similar in nature to those reached in [32] with respect to the short-term forecasting model for Germany and Poland). The minima and maxima of projections thus generated were considered the lower and upper forecast boundaries. If subsequently realized values fell within those boundaries, the corresponding error was set to zero, otherwise it was taken to be the absolute relative error of the most accurate boundary (upper or lower).

The results are presented in Table 4. Evidently, the best predictions in terms for the number of positive cases, floor, ICU and vent censuses are achieved one day in advance, and the accuracy deteriorates with the increase in the time horizon. This was to be expected. The accuracy of mortality predictions is less dependent on the time horizon, and the relationship between the former and the latter is less pronounced. This could be explained by the relative stability of the number of mortality cases and the relatively static nature of (37).

Accuracy trajectories for the number of positive cases, inpatient, ICU and vent censuses are presented in Fig 4. It can be observed that the most accurate predictions to date have been made during periods of relative "calm", i.e., those times when the infection curve followed a declining or quasi-static pattern (approximately, May–September 2020). Periods of elevated error include "regime changes" at the end of May 2020 and September 2020 to mid-January 2021. This was also to be expected given the uncertainty not captured by the moving average or polynomial extrapolation of the future transmission coefficient, $R_0$. Overall, the model appears to have "erred on the side of caution" overestimating the expected patient census while ICU and vent censuses tend to be underestimated more often, especially during the periods of "regime change".

## Practical application

The model was distributed as an Excel spreadsheet to NS executive administrative team, physician, nursing and supply chain leadership. This form of distribution allowed those interested in scenario analysis and additional forecasting specific to their line of business the flexibility to perform additional calculations independently, without the need to engage the Clinical

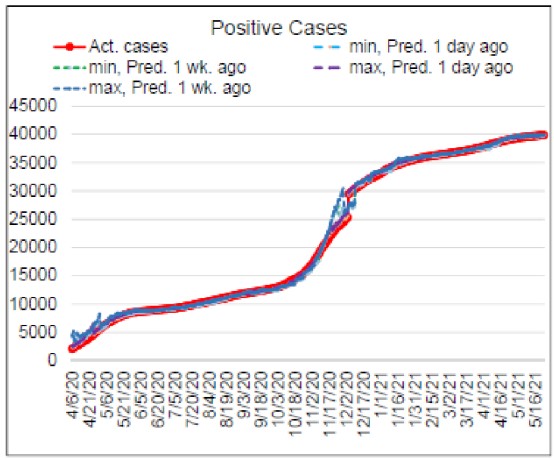

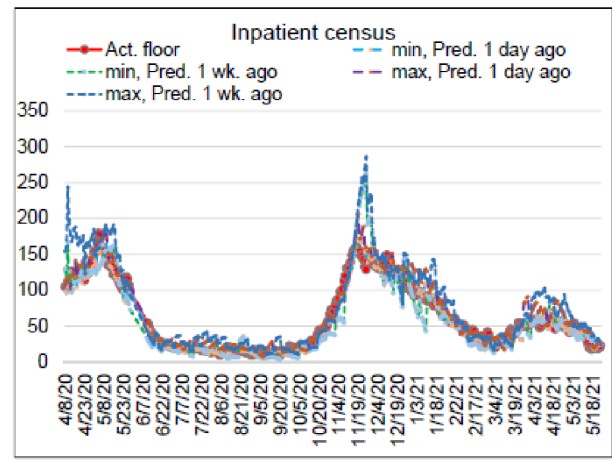

**(a)** Accuracy of the prediction for the number of NS cases

**(b)** Accuracy of the prediction for the floor census

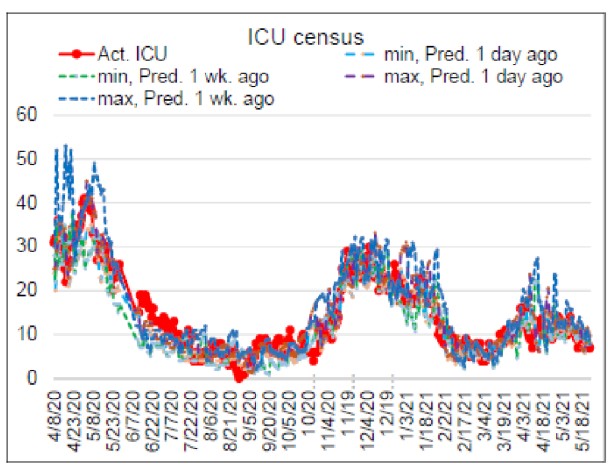

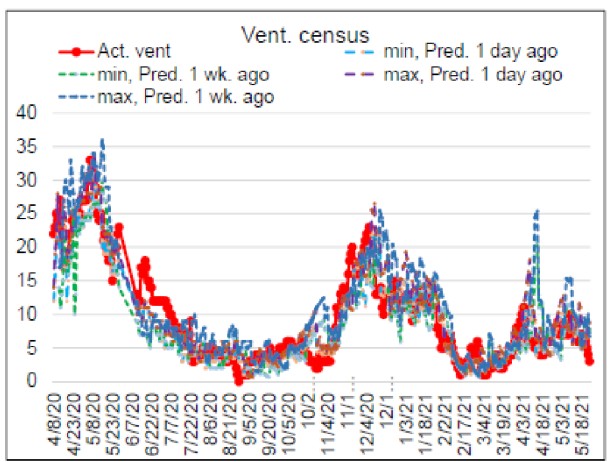

**(c)** Accuracy of the prediction for the ICU census

**(d)** Accuracy of the prediction for the vent census

**Fig 4. Historical accuracy of predictions for the number of positive NS cases, floor, ICU and vent censuses, March 10, 2020–May 24, 2021.**

Analytics team. During the initial period from Mar. 26, 2020 to June 18, 2020, the forecast was sent out daily. From June 22, 2020 to May 31, 2021, the distribution frequency was set to twice a week.

The recipients were advised to treat any outlook more than two weeks ahead of the distribution date with caution. With this caveat, the forecasts were used as a general guide for near-term future allocation of staff, hospital beds, resources and personal protective equipment. While it is difficult to quantify the impact of the model on hospital operations, according to the feedback received form stakeholders, the forecasts provided enough lead time to serve an

additional data point in making operational decisions during the most difficult times of COVID-19. According to the general feedback from forecast recipients, prediction accuracy, the level of provided detail and the timing of distribution were adequate for rendering the model useful for the purpose it was intended for.

## Conclusion

The model provided acceptable predictive accuracy for the operational stakeholders to use it as an additional data point in their decision-making process. While specifically designed for COVID-19, the algorithm used for implementing the model and distributing the results could be replicated for a similar communicable disease with high transmission, significant hospitalization, near zero reinfection and moderate mortality rates, provided sufficient requisite data describing its evolution in the community is available.

## Supporting information

**S1 Data.**
(XLSX)

**S2 Data.**
(XLSX)

## Acknowledgments

The authors would like to thank Emily Carter for thoroughly proofreading the manuscript, NorthShore University HealthSystem for the permission to use operational data in the publication of this research, our editor, Dr. Robert Jeenchen Chen and reviewers, Dr. Mahbub-Ul Alam, Dr. Ali Acar and Dr. Guillaume Beraud, for helpful comments that helped to improve this paper.

## Author Contributions

**Conceptualization:** Daniel Chertok, Chad Konchak, Nirav Shah, Kamaljit Singh, Ernest Wang, Lakshmi Halasyamani.

**Data curation:** Daniel Chertok, Chad Konchak, Nirav Shah, Kamaljit Singh, Loretta Au, Jared Hammernik, Brian Murray, Ernest Wang.

**Formal analysis:** Daniel Chertok.

**Investigation:** Daniel Chertok, Loretta Au, Ernest Wang.

**Methodology:** Daniel Chertok, Nirav Shah.

**Project administration:** Daniel Chertok, Chad Konchak, Nirav Shah.

**Resources:** Chad Konchak, Lakshmi Halasyamani.

**Software:** Daniel Chertok.

**Supervision:** Daniel Chertok, Chad Konchak, Anthony Solomonides, Lakshmi Halasyamani.

**Validation:** Daniel Chertok, Chad Konchak, Nirav Shah, Kamaljit Singh, Jared Hammernik, Brian Murray, Anthony Solomonides, Ernest Wang, Lakshmi Halasyamani.

**Visualization:** Daniel Chertok.

**Writing – original draft:** Daniel Chertok.

**Writing – review & editing:** Daniel Chertok, Chad Konchak, Loretta Au, Anthony Solomonides.

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
