## [Decision Letter · Decision Letter 0]

10 Aug 2021

PONE-D-21-23153

An operationally implementable model for predicting the effects of an infectious disease on a comprehensive regional healthcare system

PLOS ONE

Dear Dr. Chertok,

Thank you for submitting your manuscript to PLOS ONE. After careful consideration, we feel that it has merit but does not fully meet PLOS ONE’s publication criteria as it currently stands. Therefore, we invite you to submit a revised version of the manuscript that addresses the points raised during the review process.

Please revise accordingly.

We look forward to receiving your revised manuscript.

Kind regards,

Academic Editor

PLOS ONE

Journal Requirements:

3. Please include a separate caption for each figure in your manuscript

Reviewers' comments:

Reviewer's Responses to Questions

**Comments to the Author**

1. Is the manuscript technically sound, and do the data support the conclusions?

Reviewer #1: Yes

Reviewer #2: Yes

2. Has the statistical analysis been performed appropriately and rigorously? 

Reviewer #1: Yes

Reviewer #2: Yes

3. Have the authors made all data underlying the findings in their manuscript fully available?

Reviewer #1: Yes

Reviewer #2: Yes

4. Is the manuscript presented in an intelligible fashion and written in standard English?

Reviewer #1: Yes

Reviewer #2: Yes

5. Review Comments to the Author

Reviewer #1: Thank you so much for sending the draft for review. The manuscript looks in good shape for publication. Some minor suggestions:

Page 1, Line 3: spacing error in Health System

Page 1, Line 4: What is IL here, please elaborate during first use

Page 1, Line 18: Where are reference 21-24? It seems reference number 25 came after 20. Please check reference.

Page 2, Line 24: What is SIR, please elaborate during first use

Good luck with the publication.

Reviewer #2: The development of operationally applicable models to predict the effects of an infectious disease on the health system is important in combating infectious diseases. That's why I think this article has scientific significance.

6. PLOS authors have the option to publish the peer review history of their article (what does this mean?). If published, this will include your full peer review and any attached files.

Reviewer #1: **Yes: **Mahbub-Ul Alam

Reviewer #2: **Yes: **Ali Acar

---

## [Author Response · Author response to Decision Letter 0]

12 Aug 2021

Dr. Robert Jeenchen Chen, MD, MPH

Academic Editor

PLOS ONE

August 11, 2021

Dear Dr. Chen,

Thank you for an expeditious review of our article entitled “An operationally implementable model for predicting the effects of an infectious disease on a comprehensive regional healthcare system” by Daniel Chertok, Chad Konchak, Nirav Shah, Kamaljit Singh, Loretta Au, Jared Hamernik, Brian Murray, Anthony Solomonides, Ernest Wang and Lakshmi Halasyamani. On behalf of all authors, I would like to thank the reviewers, Dr. Mahbub-Ul Alam and Dr. Ali Acar for their helpful comments that improved paper content and readability. In response to these comments, we made the following changes:

1. Page 1, Line 4 spells out “Illinois”, instead of “IL”, as the name of the US state;

2. Page 2, Lines 25 – 26 (previously, Line 24) spells out “Susceptible-Infected-Recovered” instead of “SIR”;

3. References have been reorganized to appear in the order of citation in the paper. We have added missing citations for the references specified by Reviewer #1. In addition, we verified that all references have a corresponding citation. 

4. In the spirit of this comment and in order to improve readability, we converted all footnotes to inline comments, in compliance with the journal requirements. This change resulted in placing citations formerly appearing in the footnotes directly into the text of the paper.

In response to the comment from Reviewer #1 referencing Page 1, Line 3, we would like to point out that “NorthShore University HealthSystem” is the registered name of our organization, and this spelling needs to remain intact throughout the paper (please see www.northshore.org for reference).

Please note that, while the revised unmarked version of the manuscript contains citation links in proper order, the version containing tracked changes lists them in the order in which they appeared in the first submission. No references were removed or added to the revised version compared to the original.

Please do not hesitate to reach out to me with any remaining questions regarding our paper.

Sincerely,

Daniel Chertok, Ph.D.,

Senior Data Scientist – Clinical Analytics, Health Information Technology

NorthShore University HealthSystem, 

4901 Searle Parkway, Suite 1450, Skokie, IL 60077, U.S.A.

Tel.: 847-982-5226 Email: dchertok@northshore.org

---

## [Decision Letter · Decision Letter 1]

3 Sep 2021

PONE-D-21-23153R1An operationally implementable model for predicting the effects of an infectious disease on a comprehensive regional healthcare systemPLOS ONE

Dear Dr. Chertok,

Thank you for submitting your manuscript to PLOS ONE. After careful consideration, we feel that it has merit but does not fully meet PLOS ONE’s publication criteria as it currently stands. Therefore, we invite you to submit a revised version of the manuscript that addresses the points raised during the review process.

Please revise.

We look forward to receiving your revised manuscript.

Kind regards,

Academic Editor

PLOS ONE

Journal Requirements:

Reviewers' comments:

Reviewer's Responses to Questions

**Comments to the Author**

1. If the authors have adequately addressed your comments raised in a previous round of review and you feel that this manuscript is now acceptable for publication, you may indicate that here to bypass the “Comments to the Author” section, enter your conflict of interest statement in the “Confidential to Editor” section, and submit your "Accept" recommendation.

Reviewer #1: All comments have been addressed

Reviewer #3: (No Response)

2. Is the manuscript technically sound, and do the data support the conclusions?

Reviewer #1: Yes

Reviewer #3: Yes

3. Has the statistical analysis been performed appropriately and rigorously? 

Reviewer #1: Yes

Reviewer #3: Yes

4. Have the authors made all data underlying the findings in their manuscript fully available?

Reviewer #1: Yes

Reviewer #3: Yes

5. Is the manuscript presented in an intelligible fashion and written in standard English?

Reviewer #1: Yes

Reviewer #3: Yes

6. Review Comments to the Author

Reviewer #1: Thanks authors for addressing all comments. This version of the manuscript is acceptable; no further comments.

Reviewer #3: Chertok et al developed a model to predict to forecast the number of COVID patient in the hospital and made it available to the public with this article.

The previous reviewers comment have been addressed. The manuscript is well written.

However, I have some comments:

1°) In mathematical modelling, the choice of a model type is a mandatory preliminary step. Therefore a justification for the choice of a SI(R) model is necessary, beyond the "widely accepted practical approach...". In particular, why not choosing an IBM or a stochastic SIR model

2°) No other software than Excel is mentioned, however, I think authors may have used others software, in particular for bootstrapping. If this the case, it should be mentioned.

3°) Authors should explain, justify and interpret the exponential decay suggested by empirical data in Fig 1. And I'm not sure I understood where these empirical data came from.

4°) I found rather curious that the NS population (Equation 18) has to be estimated while there must others sources able to provide approximation (page 9, first paragraph of Worked example). Besides, it would be useful to find sources to assess the relevance of this assumption.

5°) Were holidays and weekend dealt in a particular way? If taken into account specifically, and even if it is not the case, is there an influence on the flow of patients?

6°) Generally speaking, I found disappointing that the discussion was focused almost only on model accuracy. Comments on the practical usefulness would useful. Comparison with others existing models would be even more useful.

7°) In the conclusion, please justify the sentence in line 269 to 272, as this comment is not really justified by the manuscript.

7. PLOS authors have the option to publish the peer review history of their article (what does this mean?). If published, this will include your full peer review and any attached files.

Reviewer #1: **Yes: **Mahbub-Ul Alam

Reviewer #3: No

---

## [Author Response · Author response to Decision Letter 1]

21 Sep 2021

Dr. Robert Jeenchen Chen, MD, MPH

Academic Editor

PLOS ONE

September 13, 2021

Dear Dr. Chen,

Thank you for a detailed review of our article entitled “An operationally implementable model for predicting the effects of an infectious disease on a comprehensive regional healthcare system” by Daniel Chertok, Chad Konchak, Nirav Shah, Kamaljit Singh, Loretta Au, Jared Hamernik, Brian Murray, Anthony Solomonides, Ernest Wang and Lakshmi Halasyamani. On behalf of all authors, I would like to thank anonymous reviewer #3 for helpful comments that improved paper content and readability. Below are our responses to these comments:

1. The choice of the SIR model was dictated primarily by the requirement of simplicity, tractability and distributability of the predictive software via an Excel spreadsheet. The use of a stochastic SIR model would have required making an assumption about the nature of the stochastic term and calibrating it to the observed infection dynamic. Likewise, IBM models introduce a level of complexity that require a substantial investment into their implementation and analysis. Dynamically adapting a model of this class to a rapidly changing situation on the ground would not be possible without more advanced computational tools and would deprive the user of the simplicity, transparency and flexibility of an Excel spreadsheet. Section “Materials and methods” of the paper has been expanded and references 5 – 9 added in order to address this concern directly in the paper.

2. No software, other than Microsoft Excel has been used in the preparation of the manuscript. Bootstrapping was performed using Excel Solver add-in called inside a VBA macro. 

3. The exponential decay described in Fig. 1 stems from the fact that NorthShore University HealthSystem was at the forefront of initial testing and detection of COVID-19. For this reason, the number of positive cases during the early stages of the pandemic was disproportionately high compared with the (estimated) share of the corresponding geographical region. As the pandemic progressed, we observed a reduction in the aforementioned proportion to a “steady-state” level reflected in Eq. 18. Section “Estimating the number of potential NS patients” has been expanded to make this argument more transparent.

4. The exact count of true NorthShore patients at any given time depends of the definition of what constitutes a NS patient, moreover, many new patients presented at NS facilities through the emergency department without having had any previous contact with NorthShore. This necessitated the need to estimate the population in the catchment area that was likely to seek treatment at NS facilities. We are not aware of any sources that could help us estimate this number in an alternative way.

5. Weekends were implicitly addressed by averaging transmission coefficient over the preceding seven days. At one point in the pandemic, we attempted to extrapolate the transmission curve using a periodic term, however, those efforts did not lead to a reliable prediction. This argument was excluded from the paper for both brevity and clarity of exposition.

6. Subsection “Practical application” was added to “Results and Discussion” to include comments on the practical usefulness of the model.

7. The argument at the end of the conclusion was rephrased to be a more generic statement referencing the features, rather than the name, of COVID-19. 

Please do not hesitate to reach out to me with any remaining questions regarding our paper.

Sincerely,

Daniel Chertok, Ph.D.,

Senior Data Scientist – Clinical Analytics, Health Information Technology

NorthShore University HealthSystem, 

4901 Searle Parkway, Suite 1450, Skokie, IL 60077, U.S.A.

Tel.: 847-982-5226 Email: dchertok@northshore.org

---

## [Decision Letter · Decision Letter 2]

5 Oct 2021

An operationally implementable model for predicting the effects of an infectious disease on a comprehensive regional healthcare system

PONE-D-21-23153R2

Dear Dr. Chertok,

We’re pleased to inform you that your manuscript has been judged scientifically suitable for publication and will be formally accepted for publication once it meets all outstanding technical requirements.

Kind regards,

Academic Editor

PLOS ONE

Additional Editor Comments (optional):

Reviewers' comments:

Reviewer's Responses to Questions

**Comments to the Author**

1. If the authors have adequately addressed your comments raised in a previous round of review and you feel that this manuscript is now acceptable for publication, you may indicate that here to bypass the “Comments to the Author” section, enter your conflict of interest statement in the “Confidential to Editor” section, and submit your "Accept" recommendation.

Reviewer #1: All comments have been addressed

Reviewer #3: All comments have been addressed

2. Is the manuscript technically sound, and do the data support the conclusions?

Reviewer #1: Yes

Reviewer #3: Yes

3. Has the statistical analysis been performed appropriately and rigorously? 

Reviewer #1: Yes

Reviewer #3: Yes

4. Have the authors made all data underlying the findings in their manuscript fully available?

Reviewer #1: Yes

Reviewer #3: Yes

5. Is the manuscript presented in an intelligible fashion and written in standard English?

Reviewer #1: Yes

Reviewer #3: Yes

6. Review Comments to the Author

Reviewer #1: Thank you so much for the revision. Authors have addressed all suggestions provided previously. The manuscript is now ready for publication.

Reviewer #3: All comments have been addressed.

I appreciate that the authors were able to explain the necessity of choosing a "simple" model in order to be able to deliver timely results, which is definitely the appropriate choice in such time.

small typo:

line 42: "was informed"

7. PLOS authors have the option to publish the peer review history of their article (what does this mean?). If published, this will include your full peer review and any attached files.

Reviewer #1: **Yes: **Mahbub-Ul Alam

Reviewer #3: **Yes: **Guillaume Beraud

---

## [Editor Report · Acceptance letter]

7 Oct 2021

PONE-D-21-23153R2 

An operationally implementable model for predicting the effects of an infectious disease on a comprehensive regional healthcare system 

Dear Dr. Chertok:

I'm pleased to inform you that your manuscript has been deemed suitable for publication in PLOS ONE. Congratulations! Your manuscript is now with our production department. 

Kind regards, 

on behalf of

Dr. Robert Jeenchen Chen 

Academic Editor

PLOS ONE